# Determining the Profile of People with Fall Risk in Community-Living Older People in Algarve Region: A Cross-Sectional, Population-Based Study

**DOI:** 10.3390/ijerph19042249

**Published:** 2022-02-16

**Authors:** Carla Guerreiro, Marta Botelho, Elia Fernández-Martínez, Ana Marreiros, Sandra Pais

**Affiliations:** 1Algarve Biomedical Center Research Institute (ABCRI), University of Algarve, 8005-139 Faro, Portugal; csguerreiro@ualg.pt (C.G.); mcbotelho@ualg.pt (M.B.); ammarreiros@ualg.pt (A.M.); spais@ualg.pt (S.P.); 2Department of Nursing, University of Huelva, 21004 Huelva, Spain; 3Department of Nursing, University of Sevilla, 41009 Sevilla, Spain; 4Faculty of Medicine and Biomedical Sciences, University of Algarve, 8005-139 Faro, Portugal; 5Comprehensive Health Research Center (CHRC), 1150-082 Lisboa, Portugal

**Keywords:** aging, risk factors, accidental falls, risk assessment

## Abstract

One in three people aged 65 years or older falls every year. Injuries associated with this event among the older population are a major cause of pain, disability, loss of functional autonomy and institutionalization. This study aimed to assess mobility and fall risk (FR) in community-living older people and to determine reliable and independent measures (health, social, environmental and risk factors) that can predict the mobility loss and FR. In total, 192 participants were included, with a mean age of 77.93 ± 8.38. FR was assessed by EASY-Care (EC) Standard 2010, the Tinetti Test and the Modified Falls Efficacy Scale (MFES). An exploratory analysis was conducted using the divisive non-hierarchical cluster method, aiming to identify a differentiator and homogeneous group of subjects (optimal group of variables) and to verify if that group shows differences in fall risk. Individually, the health, social, environmental and risk factor categories were not found to be an optimal group; they do not predict FR. The most significant predictor variables were a mix of the different categories, namely, the presence of pain, osteoarthritis (OA), and female gender. The finding of a profile that allows health professionals to be able to quickly identify people at FR will enable a reduction in injuries and fractures resulting from falls and, consequently, the associated costs.

## 1. Introduction

The increase in the number of people in the older population is a global phenomenon. In 2017, Europe had 183 million people aged 60 or older; it is estimated that, by 2050, this number will increase to 247.2 million [1].

Portugal is no exception to the European landscape, as, by 2019, the number of people aged 65 or older reached 2,262,325 people, with an aging index of 161.3, which is predicted to increase to 355.3 by 2060 [2]. In the Algarve region, the aging index in 2019 was 145.4, with it estimated to increase to 323.3 by 2060 [3].

The last several years have seen a rise in complications due to aging in the population, including a greater prevalence of chronic diseases associated with aging, reduced mobility, increased number of falls, increased morbidity and even prolonged hospitalization [4].

The degree of difficulty in walking is an important, standard measure of mobility among older people, as those who report difficulty in walking experience more health problems, lower levels of functionality and greater dependence [5,6,7,8]. As such, the less independent an older individual is, the greater the fall risk (FR) [9].

One in three people aged 65 years or older falls every year, with half of these people reporting falling more than once [10]. In addition, falls are also the leading cause of death in older people. [11]. Injuries associated with falls among the older population are a major cause of pain, disability, loss of functional autonomy and institutionalization, negatively affecting the physical, psychological, and socioeconomic status of the individual [12], which, ultimately, exerts a great impact on their quality of life [13]. Falls, therefore, represent a public health problem in older adults with high associated costs both directly and indirectly [10].

There is a need to monitor, manage, and promote screening programs for fall prevention in the older population, including personalized FR assessments. However, prediction of falls in older adults is a complicated clinical issue due to its multifactorial nature, and, therefore, several FR assessment methods are available [10]. Effective FR screening is still underutilized and not routinely integrated into clinical practice; consequently, many older adults do not undergo comprehensive FR screening, nor do they receive targeted prevention strategies [14]. Since there is no gold standard to identify FR in conjunction with its multifactorial nature [15], the main goal of this study is to determine the set of reliable and independent measures (health, social, environmental, and intrinsic risk factors) that can predict a profile of individuals with mobility loss and FR.

## 2. Materials and Methods

This study is a prospective, observational cross-sectional secondary analysis, prepared in accordance with the EQUATOR statement and the guidelines of the STROBE, with data from the study approved by the ethics committee of the University of Algarve: “Effect of physical activity on the physical and mental health of the senior population”, which were sponsored by Project: 0551_PSL_6_E–PSL–“Programme for a Long-lived Society”, INTERREG V-A Spain-Portugal (POCTEP) and financed by the cross-border Spain–Portugal European Territorial Cooperation Program and the European Regional Development Fund.

### 2.1. Participants and Setting

Sampling was done in two phases. In the first phase, the recruitment of participants was done through flyers posted at senior universities, local health care centers, senior occupational centers, municipalities, and sports complexes. In the second phase, the sampling additionally followed the concept of snowball sampling (non-random sample). The period of recruitment was between March and May 2021; the sample size corresponds to the response rate obtained in that period.

Following sampling, 30 individuals were excluded based on the pre-established exclusion criteria: the presence of severe cognitive impairment; being institutionalized and not being able to walk independently. In total, 192 independent community-living individuals over 60 years old and residing in the Algarve region were included in this study.

The included participants (141 women and 51 men) were aged 61 to 101 years (mean age 77.93 ± 8.38–woman 77.63 ± 0.68; men 78.76 ± 1.30), freely volunteered to participate and provided written informed consent.

### 2.2. Measures

To characterize the population, self-reported measures were evaluated, and data was collected via a personal interview with each participant utilizing a questionnaire created for this purpose. The measures were grouped as follows.

#### 2.2.1. Health Measures

The following health measures were evaluated: cardiovascular diseases, respiratory diseases, sensory deficits, musculoskeletal disorders, endocrine diseases, central nervous system diseases, cognitive deficits (Mini-Mental State Examination-MMSE), pain, osteoarthritis (OA), medication and nutrition (Mini Nutritional Assessment-MNA).

To assess cognitive impairment, the MMSE instrument was used with the established cutoff points previously defined for Portugal of ≤22 in those with 0–2 years of education, ≤24 in those with 3–6 years of education and ≤27 in those with ≥7 years of education [16]. 

The MNA instrument was used to determine the risk of malnutrition with cutoff points of <17–malnourished, 17–23.5–under risk of malnutrition, ≥24–normal nutritional status [17].

To define the clinical diagnosis of OA, we considered the National Institute for Health and Care Excellence (NICE) criteria and the American College of Rheumatology clinical criteria for the diagnosis of OA [18,19].

#### 2.2.2. Risk Factors

The risk factors evaluated were body mass index (BMI), smoking and falls in the last year.

#### 2.2.3. Social Measures

The social measures that we included were age, gender, marital status, household inhabitants, working status, education, and requiring a caregiver.

#### 2.2.4. Environmental Measures

To characterize the environment in which the participants were situated, the sociodemographic environment (rural, urban, and semi-urban) and housing type were collected.

#### 2.2.5. Fall Risk

To assess fall risk (FR), three different instruments were used: the EASY-Care (EC) Standard 2010–FR dimension, the Tinetti Test–Performance Oriented Mobility Assessment II (POMA II) and the Modified Falls Efficacy Scale (MFES):The EC system was developed by Ian Philip over several years [20] and has been adapted, as well as validated, for Portugal [21]. The EC–FR dimension assesses questions regarding needs and priorities that predict an increased fall risk and/or injuries resulting from falls. The questions involve aspects including difficulties and lack of safety in movement, occurrence of falls, problems with the feet, and excessive alcohol consumption. It is scored between 0–8 points, with an FR cutoff point of ≥3 points.The Tinetti Test was developed by Tinetti (1986) and was validated for the Portuguese population by Petiz (2002). It estimates the predisposition for falls in older individuals through the quantitative assessment of tasks related to mobility and balance. The test is divided into two parts (total 28 points); higher scores signify better balance. The first part assesses static balance (9 items–maximum of 16 points) and the second part assesses dynamic balance (10 items–maximum of 12 points). A score > 24 points indicates a low FR; a score of 19–24 points indicates a moderate FR; a score < 19 indicates a high FR [7,22].The MFES evaluates fear and FR in the older people in 14 daily activities within and outside of their residence with a rating of 1 to 5 (1–“Not confident”; 5–“Confident”). Higher scores represent a lower risk and fear of fall, as well as a lower functional limitation. The following cutoff points were used: 0–28 points–fear of falling and functional limitation; 29–42 points–moderate confident/minimal functional limitation; ≥43 points- without functional limitation [23].

More than one method for assessing the FR was used. Recent studies have found that different methods of assessing FR generate different proportions of participants with high/low FR, with a low level of agreement between them, as each instrument focuses more on isolated factors. These studies have shown that combining several assessment methods of FR may improve the validity of screening for FR in older individuals, since this provides a more holistic evaluation [23,24,25,26]. In this way, an FR variable was created as a positive result in 3 of the assessments. The period of data collection was between June 2021 and September 2021.

### 2.3. Statistical Analysis

Characteristics of the study population were calculated as means and standard deviation for continuous variables and frequencies (absolute and relative) for categorical variables for the entire study sample as well as when considering those who have fallen and those who have not fallen.

An exploratory analysis was completed using the divisive non-hierarchical cluster method to determine the multivariate effect of the combined and independent variables, with the aim to identify a specific profile (optimal group of variables).

The cluster analysis was based on the following algorithmic structure:(a)The study variables were identified, distributed and classified by domains (health, social, environmental and risk factors), cited in the scientific literature as fall risk factors. These variables are not included in the scoring of the falls risk assessment instrument.(b)The clustering analysis was carried out by the Two Step Cluster technique with the study variables previously identified and independent of the fall risk to understand if the participants were homogeneously grouped and whether the clusters were of good quality. Several groups of study variables were performed, until finding good quality combinations that homogeneously differentiated the participants.

For the automatic generated groups of clusters with better quality, the mean for each instrument of fall risk assessment was calculated and analyzed by inferential statistics, namely, Pearson’s chi-squared test, with a significance level of 0.05.

All analyses were conducted with IBM SPSS Statistics version 27 (IBM, New York, NY, USA). Descriptive statistics were calculated to characterize the study sample. The normality of data distribution was assessed by the Kolmogorov–Smirnov (K-S) test and parametric or non-parametric tests were used.

## 3. Results

### 3.1. Demographic and Clinical Data

A total of 89.1% of the sample population was retired, 54.7% lived in an urban environment, 40.6% lived alone and the majority reported that they did not need a caregiver (75.5%).

In relation to associated diseases, sensory deficits, pain, and cardiovascular diseases hold the highest prevalence, equating to 97.9%, 71.7% and 64.2%, respectively. OA was present in 60.9% of participants. The data characterization of each group (fall risk and non-fall risk) is presented in Table 1. 

### 3.2. Mobility and Fall Risk Data

The prevalence of the FR in our sample (with FR in the 3 tests utilized) was 33.3%.

### 3.3. Predictor’s Variables of Fall Risk and Mobility Loss

In total, 39 multivariate groups of variables were performed, 6 with health variables, 6 with social variables, 4 with environmental variables, 1 with risk factors and 22 with variables from all categories.

The health, social, environmental and risk factor groups did not reveal to be an optimal group of variables that predict FR.

Of the 22 multivariate groups from all categories, 2 groups (A and B) showed a good quality result. In these two groups, the FR showed statistically significant differences, *p* ≤ 0.05 (Table 2).

In cluster analysis A, the cluster that presented better quality and that included a higher number of participants (44.9%, *n* = 84) was cluster A1. In cluster analysis B, the cluster that presented better quality and that included a higher number of participants (34.2%, *n* = 64) was cluster B1. Both A and B found statistically significant differences for the FR (*p* = 0.017 and <0.001, respectively).

In cluster A1, the predictor variables are the presence of pain (1), OA (1) and female gender (0.93); in group B1, they are female gender (1) and requiring a caregiver (0.52). Pain, OA and female gender show a significant association with having fallen. (Table 3).

## 4. Discussion

Falls are a significant public health problem in the older population; therefore, being able to predict these is highly beneficial [27]. Unfortunately, identifying older individuals at a high risk of falls in primary health care may be difficult, as the procedure to do so is time consuming and, thus, not feasible within the constraints of daily practice [28]. Health care professionals, therefore, need a simple and pragmatic clinical approach, easily utilized in daily practice, that is able to identify older individuals who may be at high risk of falling.

There are numerous inter-related fall risk (FR) factors; the likelihood of a fall increases with the increasing number of these. It, therefore, becomes difficult to isolate a single risk factor that is responsible [29]. It is, thus, crucial to be able to identify those individuals with multiple risk factors in order to select and maximize the effectiveness of any proposed intervention [29,30]. The initial screening protocol should focus time and financial resources on those individuals at increased risk and aim to prevent any unnecessary inconvenience to those with a low risk [30].

Predictive tools based on a small number of variables could be preferable, as their use is generally easier and time efficient. However, this may result in a less accurate prediction. The need to incorporate several variables in a manner that is accessible, timely and that results in an accurate prediction must be considered when designing a feasible screening tool [30].

Multiple factors can cause falls and according to the literature review by Park, two assessment tools used together are better able to evaluate the characteristics of FR in older individuals whilst maximizing the advantages of each tool in predicting the occurrence of falls. Within this study we, therefore, used three different instruments to assess FR and to ensure a better ability in determining the predictive FR factors and the prevalence of FR.

The Tinetti test is an appropriate tool to identify FR [24], however, the large number of versions, test items, scoring and cut-off values, affect its validity and reliability [25]. The EC is a valid, comprehensive, and acceptable tool centered around the older individual’s priorities for promoting their well-being, however, there is limited evidence for its reliability and use as a population-level needs assessment [26]. Finally, the MFES is a useful measurement tool to assess confidence in daily activities and includes a scale of self-perception; however, it should be accompanied by more accurate and non-self-reported assessments [23]. Due to these limitations, we decided to use these instruments in conjunction. Additionally, the accuracy outputs of these FR assessment tools have revealed that no method stands out from the others [31].

Considering these aspects, the present study aimed to find a set of variables that take these issues into account and determine a profile of individuals with FR.

When we investigated the variables predicting falls, we found two optimal groups of variables, which, when evaluated together, highlight specific variables that significantly influenced the FR. Group A and B demonstrated that, to assess the FR, health and social variables must be evaluated. However, most of the time, only biological variables are surveyed in FR assessments [30]. The World Health Organization (WHO) points out that the risk factors result from the dynamic interaction of various risks in all categories and that this assessment must be multivariate [29].

Our results show that the set of variables most predictive of FR are having pain, osteoarthritis (OA), and female gender in group A. Having pain, OA, female gender and not having a caregiver were the set of variables most predictive of FR in group B. When we compared these variables with individuals who had fallen within the last year, being female, having pain and OA were highlighted. On the other hand, not having a caregiver was not associated with the occurrence of falls. This association further reinforces the profile of people determined by the clustering analysis. These findings may be used to build a screening tool for fall prediction.

Fall risk factors found in this study were similar to those within the current literature. Chen et al., aimed to develop an FR assessment profile in older individuals and obtained the following profile: being a woman, living alone, having urinary incontinence, self-reported poor health, pain, hospitalization in the previous year, low score in ADL and low mobility score [32]. Living alone can imply greater functional capacity; however, injuries and the outcomes thereof can be worse, particularly if the person is unable to get up off the floor [29].

Several studies indicate that women are more exposed to the occurrence of a fall event, with a probability of falling 1.51 times greater than men [32]. Women exhibit greater physical fragility and less muscle mass and strength compared to men of the same age [33,34]. Gait has also been described to be related to a higher FR in women due to the longer stride time and greater gait variability [35]. These authors indicate that the mechanisms behind this difference appear to be multifactorial and further investigations are warranted.

Concerning pain, the available literature identifies chronic pain as being strongly associated with falls and prevalent among the older adults [32]. Pain is associated with deficits in mobility and balance, as well as impaired gait; it is common in 76% of community dwellings for older people. Studies that specify the location of the pain indicate that pain is often in the hip, knee, and spine, or is described as generalized pain associated with an increased FR. However, foot and chronic pain are those that show a greater risk. This is noteworthy; these factors must be included in clinical practice [36].

A study that aimed to examine the prevalence and risk factors associated with falls concluded that female gender and chronic pain conditions, especially arthritis, were prevalent risk factors for falls [37]. Several prospective studies have shown an association between OA, the most common type of arthritis, and falls. The most widely accepted possible explanations for the increased risk of fracture in patients with lower limb OA include an increase in the rate of bone and muscle loss, joint pain, and stiffness, which, in turn, leads to increased body sway and, thus, greater propensity for the occurrence of falls and fractures [38,39,40,41]. One study showed that women after menopause who reported OA exhibit an increased FR of around 25% and that these results explain the observed increase in fractures that exists in this population [38]. Age over 65 and a diagnosis of OA increases the risk of falling 1.5 times more compared to those without OA [42].

The association between female gender, pain, and OA as predictors of FR that resulted from this study is in line with the most recent literature. Not having a caregiver is also confirmed by the literature as a contributor to a higher FR.

The Algarve region contains an older population, representative of the world panorama, in which there are more females than males. The results of this study determined the predictive FR factors in the senior Algarve population that allow for the identification of individuals with these characteristics. We believe that this finding holds implications for clinical practice in the region, as the identification of a specific profile of individual with FR will help to screen these patients within the primary health care setting in a simpler, faster and more efficient manner.

This study can, therefore, contribute to the creation of an algorithm that helps to monitor individuals with this profile and can be included in programs or strategies to prevent falls, particularly exercise programs promoting muscle strength increase and weight loss, that are directly related to osteoarthritis and associated pain. The most efficient and cost-effective fall-reduction programs involve systemic FR assessment with targeted interventions that include physical activity, medical management, and environmental inspection and hazard removal, as advised by the Arthritis Foundation [43].

The limitations of this study were as follows: some data were self-reported, which may have reduced the accuracy of the data and, therefore, resulted in memory bias. The sample showed a significantly low FR, which may have influenced the results. On the other hand, this study exhibited several strengths, namely, the risk profile developed, which was based on a large sample in relation to the size of the Algarve population and from different municipalities in the region. Additionally, the clinical interviews were conducted by an experienced and trained research group to minimize self-reporting bias.

## 5. Conclusions

This study found fall risk (FR) factors among community-dwelling older individuals in the Algarve region. The profile of individuals found to possess an FR were female, with pain and OA. This profile allows health professionals to quickly identify individuals at FR and to possibly reduce the number of injuries and fractures resulting from these falls and, consequently, the costs associated with these.

Future studies should assess the FR in a representative number of people in the Algarve region. This type of analysis should also be carried out in a sample with a high FR. In addition, it would be important to evaluate these variables utilizing a longitudinal prospective study.

## Figures and Tables

**Table 1 ijerph-19-02249-t001:** Sample characterization.

Variable		Non-Fall Risk n (%)	Fall Risk n (%)	*p* Value
Age, mean (SD)		76.79 ± 8.40	80.17 ± 7.94	0.013 ^1^
Gender	Male Female	43 (33.6) 85 (66.4)	8 (12.5) 56 (87.5)	0.002 ^2^
Marital Status	Married Single Divorced Widow	62 (48.4) 5 (3.9) 14 (10.9) 47 (36.7)	25 (39.1) 2 (3.1) 5 (7.8) 32 (50.0)	0.369 ^2^
Household Inhabitants	Alone Spouse Family Spouse and children	50 (39.1) 57 (44.5) 15 (11.7) 6 (4.7)	28 (43.8) 19 (29.7) 15 (23.4) 2 (3.1)	0.086 ^2^
Requiring a caregiver		26 (20.3)	21 (32.8)	0.058 ^2^
Education	0 years 1–6 years 7–12 years ≥13 years	10 (7.8) 84 (65.6) 25 (19.5) 9 (7.0)	8 (12.5) 46 (71.9) 8 (12.5) 2 (3.1)	0.307 ^2^
Working status	Employee Retired Unemployed	8 (6.3) 111 (85.9) 9 (7.0)	3 (4.7) 61 (95.3) 0 (0)	0.133 ^2^
Sociodemographic environment	Rural Urban Semi-urban	35 (27.3) 72 (56.3) 21 (16.4)	19 (29.7) 33 (51.6) 12 (18.8)	0.822 ^2^
Housing type	Single storey house House with elevator House without elevator	63 (49.2) 17 (13.3) 48 (37.5)	41 (65.1) 5 (7.9) 17 (27.0)	0.113 ^2^
Smoking		6 (4.7)	0 (0)	0.078 ^2^
BMI	Low weight Normal weight Overweight	13 (10.7) 54 (44.6) 54 (44.6)	5 (8.8) 28 (49.1) 24 (42.1)	0.829 ^2^
Those who have fallen in the last year		35 (27.3)	28 (43.8)	0.022 ^2^
Cardiovascular diseases		78 (60.9)	44 (71.0)	0.176 ^2^
Respiratory diseases		10 (7.9)	5 (8.1)	0.964 ^2^
Sensory deficits		124 (96.9)	62 (100)	0.159 ^2^
Musculoskeletal disorders		15 (11.9)	16 (25.4)	0.018 ^2^
Endocrine diseases		37 (29.4)	19 (30.6)	0.857 ^2^
Central nervous system diseases		36 (28.1)	27 (42.9)	0.042 ^2^
Cognitive deficit		12 (9.4)	7 (10.9)	0.733 ^2^
Pain		85 (67.5)	49 (80.3)	0.067 ^2^
Osteoarthritis		70 (54.7)	47 (73.4)	0.012 ^2^
Polymedicated		31 (24.4)	16 (25.4)	0.882 ^2^
Malnutrition risk		18 (15.4)	15 (26.8)	0.074 ^2^

^1^ Mann–Whitney test. ^2^ Pearson’s Chi-squared test.

**Table 2 ijerph-19-02249-t002:** Cluster analysis.

Cluster			p-Value
Group A	Quality Clusters	Good Silhouette (1) 7	Fall Risk
	Clusters Size		
	A1 A2 A3 A4 A5 A6 A7	44.9% (n = 84) 12.8% (n = 24) 10.7% (n = 20) 9.1% (n = 17) 8.6% (n = 16) 7.0% (n = 13) 7.0% (n = 13)	0.017 ^1^
Group B	Quality Clusters	Good Silhouette (0.6) 4	
	Clusters Size		
	B1 B2 B3 B4	34.2% (n = 64) 26.2% (n = 49) 23.5% (n = 44) 16.0% (n = 30)	<0.001 ^1^

^1^ Pearson’s Chi-squared test.

**Table 3 ijerph-19-02249-t003:** Clusters fall risk predictors and association with falls.

Variables		Category Frequency (%)	Predictive Power Cluster A	Predictive Power Cluster B	Falls	*p*-Value
Health	Pain Osteoarthritis	Yes (100) Yes (100)	1 1	0.29 0.47	81% (*n* = 51) 74.6% (*n* = 47)	0.001 ^1^ 0.007 ^1^
Social	Gender Caregiving	Female (100) No (100)	0.93 -	1 0.52	85.7% (*n* = 54) 76.2% (*n* = 46)	0.007 ^1^ 0.880 ^1^

^1^ Pearson’s Chi-squared test.

## Data Availability

Data is not available due to privacy restrictions.

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
