# Peer review of "Determining the Profile of People with Fall Risk in Community-Living Older People in Algarve Region: A Cross-Sectional, Population-Based Study"

_ijerph, 2022, doi:10.3390/ijerph19042249_

Round 1

Reviewer 1 Report

The purpose of current study was ‘to assess mobility and fall risk (FR) in community-living older people and determine reliable measures (health, social, environmental and risk factors) that can predict the mobility loss and FR’. They presented interesting results based on their cluster analysis. However, there are several limitations in the current study to be published as an original study.

  1. Why did they perform cluster analysis for this study with small cases? They should clearly present the reason why they perform cluster analysis in the method section. I think conventional logistic model rather than cluster analysis is more suitable for this study.

  1. They did not sufficiently present their method of cluster analysis. For example, how did they decide the number of clusters (from Table 3, I guess that they may use the silhouette method). In addition, they did not show sufficient baseline data according to the clusters.

.

  1. I think Table 1 should be revised. If the purpose of current study was to assess the fall risk, they should compare their variables between the patients with and without the fall risk.

  1. In the method, section, they stated, ‘More than one method for assessing the FR was used, once recent studies have shown that different methods of assessing the risk of falling generate different proportions of participants with high/low FR, evidencing a low agreement output between them’ Then, why were subjects considered to have a fall risk if in all 3 tests the the participant showed a FR? If the reasons are not clear, additional sensitivity analysis are required according to the different definitions of fall risk.

  1. The most important limitation of this study is that they did not investigate actual fall. The risk of fall does not mean the actual fall.

Author Response

The response to the reviewer is attached in word document

Reviewer 2 Report

General comments

It is an exciting material investigating factors associated with falls in older people. Considering that, I would recommend that:

- Authors revise the manuscript following STROBE recommendation (von Elm et al., 2007). A detailed review should include:

- Rewrite the introduction, shortening it, giving more attention to the determinants of falls in older people and the impact of falls on older´s people health and life quality. At the end of this section, should be clary presented the study goal;  

- Materials and method section needs to present a better description of study design, measurements, variables, and statistical analytical procedures used as recommended by EQUATOR statement;

- Results and conclusion needs incorporate the incorporated aspects;

Reference:

von Elm E, Altman DG, Egger M, Pocock SJ, Gotzsche PC, Vandenbroucke JP. The Strengthening the Reporting of Observational Studies in Epidemiology (STROBE) Statement: guidelines for reporting observational studies. Ann Intern Med. 2007; 147(8):573-577. PMID: 17938396

Author Response

(The authors gave the same response as above.)
